# StraitFlux - Precise computations of Water Strait fluxes on various Modelling Grids

Susanna Winkelbauer[1,2], Michael Mayer[1,2,3], and Leopold Haimberger[1,2]

[1]Department of Meteorology and Geophysics, University of Vienna, Vienna Austria
[2]b.geos, Korneuburg, Austria
[3]European Centre for Medium-Range Weather Forecasts, Bonn, Germany

**Correspondence:** Susanna Winkelbauer (susanna.winkelbauer@univie.ac.at)

**Abstract.** Oceanic transports shape the global climate, but the evaluation and validation of this key quantity based on reanalysis and model data is complicated by the distortion of the used curvilinear ocean model grids towards their displaced north poles. Combined with the large number of different grid types, this has made the exact calculation of oceanic transports a challenging and time-consuming task. Use of data interpolated to standard latitude/longitude grids is not an option since transports computed from interpolated velocity fields are not mass consistent. We present two methods for transport calculations on grids with variously shifted north poles, different orientations, and different Arakawa partitions. The first method calculates net transports through arbitrary sections using line integrals, while the second method generates cross-sections of the vertical-horizontal planes of these sections using vector projection algorithms. Apart from the input data on the original model grids the user only needs to specify the start and end points of the required section to get the net transports (for the first method) and their cross-sections (for the second method). Integration of the cross-sections along their depth and horizontal extent yields net transports in very good quantitative agreement with the line integration method. This allows to calculate oceanic fluxes through almost arbitrary sections, to compare them with observed oceanic volume and energy transports at available sections such as the RAPID array or at Fram strait and other Arctic gateways, or to compare them amongst reanalyses and to model integrations from the Coupled Model Intercomparison Projects (CMIP).

We implemented our methods in a Python package called StraitFlux. This paper represents its scienfic documentation and demonstrates its application on outputs of multiple CMIP6 models and several ocean reanalyses. We also analyse the robustness and computational performance of the tools as well as the uncertainties of the results. The package is available on github and zenodo and can be installed using pypi.

# 1 Introduction

Oceanic transports of heat, volume, and salt are integral components of the Earth's energy and mass budgets, playing a key role in regulating the Earth's climate. For instance, the Atlantic Meridional Overturning Circulation (AMOC) plays a crucial role heating the North Atlantic by transporting warm surface waters from the tropics to the North Atlantic via the Gulf Stream and cold, dense waters southward at depth. It influences the weather and climate of eastern North America and western Europe (Jackson et al., 2015), and subsequently also affects the Arctic climate and sea ice (Liu and Fedorov, 2022; Mahajan et al., 2011). A weakening of the AMOC has been reported (Caesar et al., 2018; Mayer et al., 2023a; Rahmstorf et al., 2015) over recent decades and a potential future collapse of the AMOC (Ditlevsen and Ditlevsen, 2023) would have major effects on the North Atlantic region and beyond. Monitoring oceanic currents is therefore particularly important in today's rapidly changing climate.

There exist several mooring arrays and other measurement devices capable of recording deep water velocities and other sea state variables in the oceans. For example, there are mooring lines in the Arctic gateways (Tsubouchi et al., 2012, 2018) or the Rapid Climate Change–Meridional Overturning Circulation and Heatflux Array (RAPID-MOCHA, e.g., Rayner et al., 2011) and the Overturning in the Subpolar North Atlantic Program (OSNAP, Lozier et al., 2017) for measuring the AMOC. It is desirable to compare transports calculated from those instruments with ocean reanalyses and climate models. This is challenging because the moorings are not aligned with the model grids, and the grids of ocean models, particularly in the Arctic, are complicated. A tool that facilitates a consistent comparison of flux estimates from this growing set of sources is therefore needed.

The convergence of meridians towards the North Pole pose challenges in ocean modelling. Murray (1996) proposed several global orthogonal curvilinear grids where the North Pole is placed over land areas in order to avoid singularities over the ocean. Those ideas were picked up by many modelling centres and are now commonly used in the world of ocean modelling. Fig. 1 shows examples of the two most common grid types. Many ocean models use tripolar grids, where two mesh north poles are placed over North America and Eurasia, whereat the exact location of those two North Poles varies between models. Ellipses around those poles and their normals create the new grid-lines (Madec and Imbard, 1996), which are strongly displaced in the northern latitudes when compared to a regular dipolar grid. The second main grid type is the displaced dipolar grid, where the North Pole is displaced to somewhere over land areas, usually Greenland. Hereby especially the gridlines in the proximity of the artifical pole feature a strong distortion. While solving the numerical problem of a singularity over the ocean, those curvilinear grids complicate the calculation of oceanic transports, especially in the proximity of the poles, as velocities in the direction of the artificial poles do not point in the direction of the true north and artificial zonal velocities do not point to the true east. The exact position of the poles, the angle between the native grid-lines and regular longitude-latitude lines, as well as the horizontal and vertical resolution varies between different models, forming a vast amount of different grid types that complicate inter-comparison between different models and to observations.

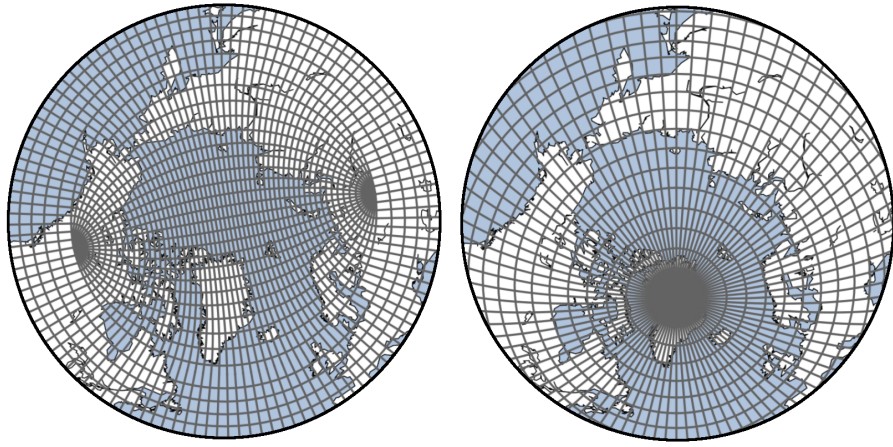

**Figure 1.** Examples of two curvilinear grids typically used for ocean modelling. Left: a tripolar grid with 2 northern poles (1 over Eurasia, 1 over North America); Right: a displaced dipolar grid with one northern pole displaced over Greenland.

For optimal accuracy and consistency the transports have to be calculated on the native grids. Horizontal interpolation of vector quantities (u,v) onto more convenient regular grids prior to the transport calculation compromises the conservation properties of the respective models, potentially leading to spurious effects and misleading results (an example is provided in Sect. 3). Finding the nearest points of the model grid has so far been done mostly manually for selected straits (e.g., Heuzé et al., 2023).

This is time consuming and becomes increasingly impractical when dealing with multiple models and multiple straits and increasing model resolution.

We have developed two methods for calculating oceanic transports on arbitrary oceanic sections, independent of grid pole placement, orientation and Arakawa partition. The first method, using line integration, yields net transports of volume, heat, salt, and ice across defined straits. The second method employs vector projection algorithms to generate cross-sections of

60 currents, temperature, and salinity in the vertical plane. We will refer to the methods as Line Integration Method (LM) and Vector Projection Method (VPM). For both methods the tedious point selection process is fully automatized. We tested our methods on various tri- and dipolar grids from the Climate Model Intercomparison Project Phase 6 (CMIP6, Eyring et al. (2016)) and show some exemplary results in Sect. 3.1.

This paper is structured as follows: In Sect. 2, we describe the fundamental concepts of both calculation methods and their

mathematical foundation. Furthermore, in Sect. 2.2, we describe the implementation of the methods in the open-source Python package StraitFlux. Sect. 3 assesses the robustness of the tools, examines their accuracy, provides application examples, and analyzes their computational performance. The final section outlines the strengths and weaknesses of StraitFlux and draws conclusions regarding its utility.

## 2 Methods

 ### 2.1 Mathematics/General idea

The general idea is to calculate oceanic volume, heat, salinity and ice transports across any chosen vertical section, typically a straight section between two land masses. We define the oceanic transports of volume (OVT), heat (OHT), salinity (OST) and ice (OIT) through a given strait as follows:

$$OVT = \int\limits_{x_1}^{x_2} \int\limits_{0}^{z_b(x)} \boldsymbol{v}_o(x,z) \cdot \boldsymbol{n} \, dz \, dx \tag{1}$$

$$OST = \int\limits_{x_1}^{x_2} \int\limits_{0}^{z_b(x)} S\boldsymbol{v}_o(x,z) \cdot \boldsymbol{n} \, dz \, dx \tag{2}$$

$$OHT = c_p \rho \int\limits_{x_1}^{x_2} \int\limits_{0}^{z_b(x)} (\theta(x,z) - \theta_{ref})\boldsymbol{v}_o(x,z) \cdot \boldsymbol{n} \, dz \, dx \tag{3}$$

$$OIT = \int\limits_{x_1}^{x_2} d(x)\boldsymbol{v}_i(x) \cdot \boldsymbol{n} \, dx \tag{4}$$

where $\boldsymbol{v}_o$ ($\boldsymbol{v}_i$) represents the velocity vector of liquid water (sea ice) and $\boldsymbol{n}$ is the vector normal to the strait - therefore their product gives velocities normal to the considered coast-to-coast section. Further, $x$ defines the along-strait extent and $z$ its depth. The boundaries $z_b$, $x_1$ , $x_2$ have to be chosen such that no water can "escape" the desired coast-to-coast section. This can be ensured if $x_e$ and $x_w$ are land points and the auxiliary fields describing model ocean depths are used appropriately.

It is also possible to calculate transports between two water points, however results should be viewed with caution and their meaningfulness is left to the discretion of the user. $S$ is the sea water salinity, $c_p$ the specific heat of seawater, $\rho$ the density of seawater and $\theta$ the potential temperature. Throughout this study, we will use $\theta$ and $T$ both synonymously for the potential temperature of seawater. For the validation in Sect. 3, $c_p$ and $\rho$ are set to 3996 Jkg$^{-1}$K$^{-1}$ and 1026 kgm$^{-3}$, respectively, as default values. However it is easy to adapt those values to individual model needs. Previous studies (Schauer and Losch, 2019;

Schauer and Beszczynska-Möller, 2009) correctly point out that true heat transports would actually demand closed volume transports through the examined straits. This is generally not the case for partial sections as transports may be compensated by flows through other passages and unbalanced volume transports would generally introduce the dependence of heat transports on the chosen temperature scale via $\theta_{ref}$. However, the "heat fluxes" as defined above are commonly used to ensure comparability

with transports derived from observations. Therefore, we will further refer to heat transports when calculating "heat fluxes" as defined here. Additionally each model's heat flux should be computed relative to a reference temperature $\theta_{ref}$, representing the mean temperature of the assessed flow. The validation part of this paper focuses on Arctic straits, therefore, we follow e.g. Heuzé et al. (2023); Muilwijk et al. (2018) and choose a universal reference temperature of $0°$C. Generally, in StraitFlux, reference temperatures are set per default to $0°$C, but may be changed to individual needs. Furthermore, following the approach of Schauer and Losch (2019) and Heuzé et al. (2023), salinity fluxes are calculated instead of freshwater fluxes to avoid the need for a reference salinity, which would vary for each model. This simplification facilitates model comparisons.

### 2.1.1 Line Integration Method (LM)

The basic principle of LM is outlined in Fig. 2a. A closed line is generated following the strait (red line) as closely as possible by connecting the faces of the individual grid cells (blue line). For an Arakawa-C grid (see inset of 2a for definition), the $u$ and $v$ components are positioned exactly at the faces of the grid cells. This positioning allows for a straightforward integration of the $u/v$ components along the meridional/zonal width of the grid cell and its depth. The sum of all grid cells vertically and horizontally then provides the net transport through the strait. In the case of an Arakawa-B grid, however, the $u$ and $v$ components are defined at the edges of the grid cells and must be transformed to the faces of the tracer grid cells. To obtain u and v velocities at the faces of the tracer cells, we adapt equations 6.46 and 6.50 from Griffies et al. (2004) (in accordance we also use the same multi-letter variable names). The $u$ velocities at eastern faces of the tracer cells ($uet$) are calculated as:

$$uet_{i,j,k} = \frac{dyu_j \, dhu_j \, u_j + dyu_{j-1} \, dhu_{j-1} \, u_{j-1}}{2 \, dyet_{i,j}} \tag{5}$$

with $dyu_j$ the meridional width of the $u/v$ cell, $dhu_j$ the depth of the u/v cell and $dyet_{i,j}$ the meridional width of the tracer cell's east side. The $v$ velocities at northern faces of the tracer cells ($vnt$) are calculated as:

$$vnt_{i,j,k} = \frac{dxu_i \, dhu_i \, v_i + dxu_{i-1} \, dhu_{i-1} \, v_{i-1}}{2 \, dxnt_{i,j}} \tag{6}$$

with $dxu_j$ the zonal width of the $u/v$ cell, $dhv_j$ the depth of the $u/v$ cell and $dxnt_{i,j}$ the zonal width of the tracer cell's north side. After transformation, transports are calculated identically as for the Arakawa-C grid. For the case of an Arakawa-A cell, where $T$, $u$ and $v$ are placed in the middle of the grid cell, we implement a similar method as for Arakawa-B and move the $u$ and $v$ components onto the cell faces. Note that while volume transports are calculated without any further use of interpolation, for heat and salinity transports, the scalar quantities of $T$ and $S$ have to be interpolated to the faces of the tracer grid cells. This is done using linear interpolation (similar to Madec, 2016, Sect. 12.3.1).

### 2.1.2 Vector Projection Method (VPM)

The second method uses simple vector projection algorithms to obtain the share of the $u$ and $v$ components that passes orthogonally through the strait. Fig. 2b shows a schematic of the VPM. For every grid cell touching the strait and their neighboring

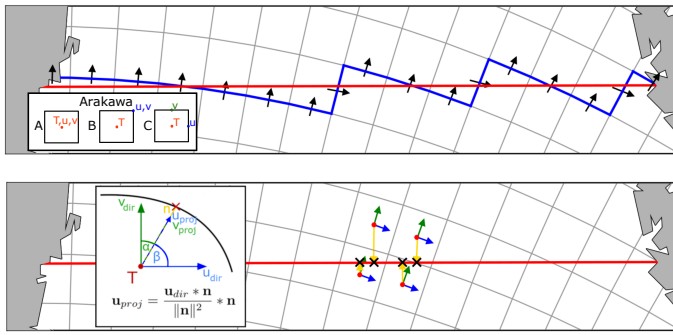

**Figure 2.** a) Schematic of the Line Integration Method, with the reference line (bold red line), the closed line generated on the native model grid (blue line) and used $u$ and $v$ components (black arrows). b) Schematic of the Vector Projection Method, the direct $u$ and $v$ vectors (blue and green) are projected onto the normal vectors (yellow) to find the portions of $u/v$ that pass orthogonally through the strait (red line).

cells (needed for the interpolation onto the strait) we calculate direction vectors of the $u$ and $v$ components (blue and green arrows), and normal vectors pointing from the tracer grid cell in the direction of the strait (yellow arrows). Then, using equations
7, the $u$ and $v$ vectors are projected onto the normal vector:

$$\boldsymbol{u}_{proj} = \frac{\boldsymbol{u}_{dir} \cdot \boldsymbol{n}}{\|\boldsymbol{n}\|^2} \cdot \boldsymbol{n}; \qquad \boldsymbol{v}_{proj} = \frac{\boldsymbol{v}_{dir} \cdot \boldsymbol{n}}{\|\boldsymbol{n}\|^2} \cdot \boldsymbol{n} \qquad (7)$$

where $\boldsymbol{u}_{dir}$ and $\boldsymbol{v}_{dir}$ represent the direction vectors of the $u$ and $v$ components, $\boldsymbol{n}$ the normal vector between tracer grid cell and strait and $\boldsymbol{u}_{proj}$ and $\boldsymbol{v}_{proj}$ the projection vectors of $\boldsymbol{u}_{dir}$ and $\boldsymbol{v}_{dir}$, passing orthogonally through the strait. The closer the angle between direction vector and normal vector ($\alpha$ and $\beta$ in Fig. 2b) to 0 (90) or 180 (270), the larger (smaller)
the amount that actually passes through the strait. Using the magnitudes of the projection vectors we calculate the $u$ and $v$ components pointing orthogonally onto the strait at all grid cells touching the strait and their neighboring cells (needed for the bilinear interpolation). Then, we multiply them with the respective vertical cell thicknesses at the u/v points and interpolate those orthogonal "transports" bilinearly onto the closest points on the reference line (black crosses in Fig. 2b, called $T_{proj}$ henceforth). In a final step we divide the interpolated "transports" by the vertical thickness of the cells on the reference line
to obtain velocities again. This results in velocity cross-sections of the vertical plane which are spaced irregularly along the along-strait distance ($x$) in accordance with the distribution of $T_{proj}$ points. The interpolation onto evenly distributed points on the section, to e.g. enable the calculation of differences with other models/reanalyses, is initially left to the user and eventually will be included in a future version of StraitFlux. By calculating transports (scalar quantities) prior to the interpolation onto the strait we ensure that the conservation properties of the models are maintained. This ensures that integration of the cross-sections
along the along-strait distance ($x$) and depth ($z$) provides net transports, which agree very well with the values obtained by the

LM (see Sect. 3). To obtain heat and salinity transports the velocity cross-sections simply have to be multiplied with the $T$ and $S$ cross-sections, which are obtained using bilinear interpolation, before integration.

## 2.2 Software Implementation

In this section, we describe the implementation of the transport calculation tools into the open-source python package Strait-
Flux. We provide an overview of the code structure and its usage.

### 2.2.1 Preprocessing

The script is designed to work with ocean reanalyses and various CMIP6 models, which may differ in terms of grid orientation, partitioning, coordinate names, units, and dimension names. Therefore, prior to the transport calculations, data preprocessing is conducted to standardize these attributes. This includes ensuring consistency in dimension names (x, y, lev, bnds), coordinate
names (lon, lat), units (SI), and the shape of longitude and latitude coordinates (2D). For CMIP6 models, preprocessing is carried out using selected tools from the open-source python package `xMIP` (https://cmip6-preprocessing.readthedocs.io/en/latest/). Ocean reanalyses are treated in a similar fashion by adapting some of the `xMIP` tools. There is no regridding involved.

To integrate transports along cell faces and depths accurately, precise information about the cell extents is required. As horizontal grid metrics are not always provided by CMIP6 models, the script automatically determines the zonal and meridional
extents using the `calc_dxdy` function. The vertical extent has to be specified prior to the calculation process. For instance, for CMIP6 the variable "cell thickness" (*thkcello*), which is available through the Earth System Grid Federation (ESGF) website (https://esgf-node.llnl.gov/search/cmip6/) for most models, may be used. Since the cell thickness is not available for all models, the function `calc_dz` enables the calculation of cell thicknesses by supplying the variable "total ocean depth" (*deptho*, also available via ESGF), and the vertical level boundaries (*lev_bnds* for CMIP6, contained in every three dimensional ocean
variable). If *deptho* is not available it is also possible to supply it from another model, preferably one with a similar grid, and the variable will automatically be interpolated to the needed grid. However, especially volume transport calculations are very sensitive to the exact ocean bathymetry used by the model. Therefore, if possible, it is advised to supply the exact fields (e.g. *thkcello* for CMIP6) and not necessarily to rely on `calc_dz`.

### 2.2.2 Index extraction

The determination of section positions for transport calculations is accomplished in the `def_indices` function. Users can specify the start and end points of a section using the `coords` parameter in the `transports` function. The section

will then follow the shortest distance along the sphere. Alternatively, users may pass specific coordinates by setting the `set_latlon=True` parameter and providing a list of latitude (`lat_p`) and longitude (`lon_p`) points. The latter option also allows the calculation of "kinked" sections. Using the 'coords' option the function generates a reference line ($ref_{line}$) con-

170 sisting of equally spaced latitude-longitude pairs whereat the interval between points on the reference line is set automatically to be suitable for the resolution of the model. When passing coordinates via the `set_latlon=True` option we advise the user to use intervals not larger than 0.4 times the resolution of the model (e.g., intervals of 0.1° for models with a resolution of about 0.25°) as coarser intervals might lead to the skipping of grid points and generate broken lines. Providing coordinates at high resolution might create duplicates in the indices found, however those will be removed automatically.

In order to calculate the net transports via line integration, a polyline along the edges of grid boxes has to be generated following the $ref_{line}$ as closely as possible (red line in Fig. 2). The function `check_availability_indices` determines the indices of the u and v components along the closed integration line:

The first point of the integration line is found by selecting the nearest point on the native grid (center of a cell) within a selection window with the size of 2 degrees around the first point on the reference curve - the size of the selection window was

180 chosen to work properly on grids with a resolution of 1 degree and higher, if needed the window size can be adapted to coarser resolutions. The subsequent points on the reference line are then found iteratively using the `select_points` function. The basic principle of `select_points` is outlined in Fig. 3. The starting point at any iteration $i$ is defined as coordinate pair [$x_s$, $y_s$]. The next point is found by comparing the distances of the four surrounding points ([$x_s$, $y_{s+1}$],[$x_{s+1}$, $y_s$],[$x_s$, $y_{s-1}$],[$x_{s-1}$, $y_s$], marked in blue), to the next point $i + 1$ on the $ref_{line}$ (red line). The point closest to the reference curve becomes the next

point on the model grid that will be included in the integration line (outlined in blue, Fig. 3a) and the new starting point [$x_s$, $y_s$] (Fig. 3b) to find the subsequent point. This is done for all points on the reference line and results in a closed sequence of grid cells along the reference line (filled blue cells). To prevent water from flowing around the strait, we advise the user to place the first and last point of the defined section over land. Transports may also be calculated for sections between ocean points. However, as the position of currents might differ between models this could lead to currents circulating around the strait and

result in significantly different results. Therefore, should the strait start/end in water a warning will be given to the user and transports should be treated with caution. The user is provided with figures of the selected line and the model land-sea mask, which can be used to check the position and length of the desired strait.

To determine whether the $u$ or $v$ component of each grid cell has to be taken, we look at the positioning relative to the previous cell (see Fig. 3c). When coming from left[1] or right to the new grid cell, the $v$ component of the cell is taken. When coming

from above/below to the new cell, the $u$ component of the previous/current cell is taken. For instance, to get to cell $j$ in Fig. 3c, we came from left, hence the $v$ component of cell $j$ is taken. In order to get to cell $j + 1$ we come from below, therefore the $u$ component of cell $j + 1$ is taken.

---

[1]Note that we follow the native grid points (x,y) and the local direction of x is not necessarily west-east and the local direction of y not south-north. For instance, coming from left here means coming from point [xi-1,yi] to point [xi,yi].

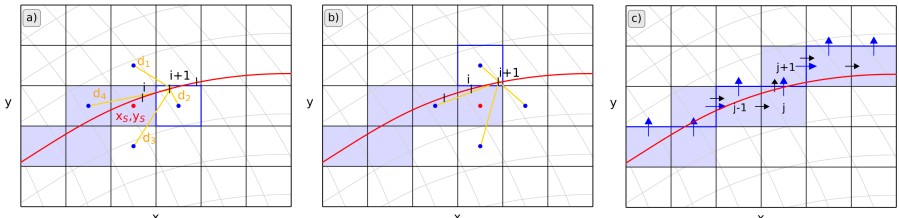

**Figure 3.** Illustration of the indices selection process using `select_points`. Lines of constant latitude/longitude are shown in grey. a) and b) Determination of consecutive grid-cells on the native grid. Distances (orange lines) of the 4 grid-cells (blue dots) surrounding the current grid-cell (red dot) are compared for all equally spaced points i along the reference line. c) Specification whether a $u$ or $v$ component should be taken. Black arrows show movement from one grid cell to the next, blue arrows show the chosen $u$ or $v$ component.

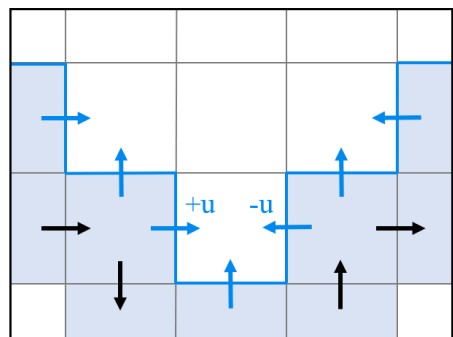

**Figure 4.** Illustration of the specification of the sign for the $u$ component. Black arrows show the direction of integration along the reference line (blue line). Blue arrows show the direction of the used $u$ and $v$ components. When coming from above (i.e. from a higher index in y) $u$ is counted positive and negative when coming from below.

Depending on the orientation of the reference curve relative to the distorted grid lines, the sign of the $u$ component may vary - this is illustrated in Fig. 4: $u$ is counted positive when cell $j+1$ is below cell $j$ (=coming from above, i.e. from a higher index in y direction) and negative when $j+1$ is above $j$ (=coming from below, i.e. from a lower index in y direction). In some cases also the sign of the $v$ component may change to negative in positive y direction, especially due to the strong bend of dipolar grids in the proximity of Greenland. Fig. 5 shows the change of the $v$ sign for Fram strait on the dipolar grid as used by the POP2 ocean model (Smith et al., 2010) in the CMIP6 model SAM0-UNICON (Park et al., 2019). If the new cell is reached by coming from the left, the $v$ component if positive (lower part of the strait in Fig. 5), and when coming from the right the whole grid is oriented upside-down and $v$ is negative (upper part of the strait in Fig. 5). Straits may be defined from west to east as well as from east to west (also north to south and south to north) in the `def_indices` function and integrated transports are defined to be positive pointing to the left of the direction of the strait. Therefore, transports for straits defined from north-west to south-east will be positive towards the north-east and negative towards the south-west.

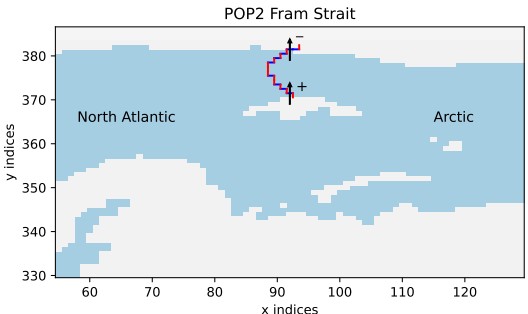

**Figure 5.** Change in sign of the $v$ component at Fram strait in the CMIP6 model SAM0-UNICON (Park et al., 2019), using a displaced dipolar POP2 ocean model. Blue lines indicate cells where $v$ components are used and red lines where $u$ components are used. In the eastern (lower in y-direction) part of the strait positive $v$ components are tilted from the strait toward the true north and are therefore counted positive, in the upper part of the strait $v$ components are tilted from the strait toward the true south and are therefore counted negative.

### 2.2.3   North fold boundary and periodic cells

Due to the periodicity of the Earth, the model domain boundaries pose a further challenge for the transport computations. Concerning the eastern and western boundaries, most ocean models have various options for dealing with periodicity. The most common options are cyclic east-west boundaries and closed boundaries. For the cyclic boundaries the values of the last 1-2 columns are set to the values of the first 1-2 columns - therefore, whatever flows out of the western end of the basin enters the eastern end and vice-versa (Madec, 2016). For the closed boundary conditions solid walls are enforced at all model boundaries

and the first and last columns are set to zero. Same is true for the north-south boundary conditions. For our application the displaced dipolar model grids are uncomplicated concerning the north-south conditions. As both poles are placed over land the southern and northern most grid-cells consist of land areas only and no oceanic transports pass the northern boundary - the grids are topologically equivalent to a cylinder (periodic in x but not in y, Smith et al. (2010)). The tripolar grids however require additional consideration along the northern boundary, as the ocean is divided by the line between the two northern grid

poles. For instance, the three-polar ORCA grid (used in multiple CMIP6 models) uses a North fold boundary with a T-point pivot (see Sect. 8.2.2 in Madec (2016)), where the upper three grid-cells are duplicated and pivoted around the line connecting the two north poles. For more information on the model specific boundary conditions see e.g., Madec (2016); Griffies et al. (2004); Smith et al. (2010).

These conditions have to be handled with care, as especially the volume transport calculation is very sensitive and can yield

useless results when there is a gap in the integration line or if any grid-cells are counted twice. StraitFlux automatically checks for overlapping cyclic boundary points and drops any duplicates. This should ensure correct transport calculations across the zonal boundaries independent of how the models deal with periodicity. Similarly, concerning the north boundary conditions, StraitFlux automatically selects the correct indices and avoids gaps and/or duplicates. We tested this successfully

for an arbitrary line going over the top boundary of the model grids for various CMIP6 models with different boundary conditions (CMCC-CM2-SR5, EC-Earth3, CanESM5, ACCESS-CM2, CAMS-CSM1-0, IPSL-CM6A-LR). Fig. A1 in the appendix shows an example for the CMCC-CM2-SR5 model. StraitFlux correctly chooses the indices so that a continuous line without overlaps is formed. While the indices selection worked for the tested models, the generated indices should still be checked to ensure a continuous line also for more complicated boundary conditions. Therefore, the code automatically outputs the warning "Attention: Strait crossing the northern boundary – make sure correct indices are chosen!" when moving across the boundary of the grid.

### 2.2.4 Line Integration Method (LM)

The actual oceanic transports are calculated through the `transports` function by integrating the velocities at the chosen $u$ and $v$ indices over the cells zonal or meridional extents and their actual vertical extent. See Sect. 2.2.1 for the computation of the needed mesh files.

For heat transports the indexed cells are additionally multiplied by the ocean's potential temperature prior to integration and for salinity transports with the cells salinity. These are defined at grid cell midpoints for Arakawa B and C grids. Therefore, the fields of $T$ and $S$ have to be interpolated to the faces of the grid cells first, which is done using linear interpolation (similar to Madec, 2016, Sect. 12.3.1). Furthermore heat/salinity transports have to be multiplied by specific heat and density (see equation 3). Those are set as constant to 3996 Jkg$^{-1}$K$^{-1}$ and 1026 kgm$^{-3}$ per default, however it is possible to adapt them to individual needs.

Transports may be calculated for longer time periods at once, where the time period may be set by the 'time_start' and 'time_end' arguments in `transports`. The final result of `transports` gives net integrated transports through the strait with the coordinate time, which are saved per default as netCDF file. The used indices and values through the cell faces can be saved on request.

### 2.2.5 Velocity Projections

The LM has good conservation properties but since the faces of the polyline can point into very different directions, it is difficult to plot cross-sections. Therefore a second method for calculating the cross-strait transports at points on the reference line has been developed. The direction of the normal vectors thus changes smoothly and allows the calculation of horizontally and vertically resolved contributions to the total transport through a respective strait.

As for the LM, the first step is to find the closest points on the native grids to the reference line. The selection of the indices proceeds similar as for LM. However, herein additionally to the closest points to the reference line also the four immediate

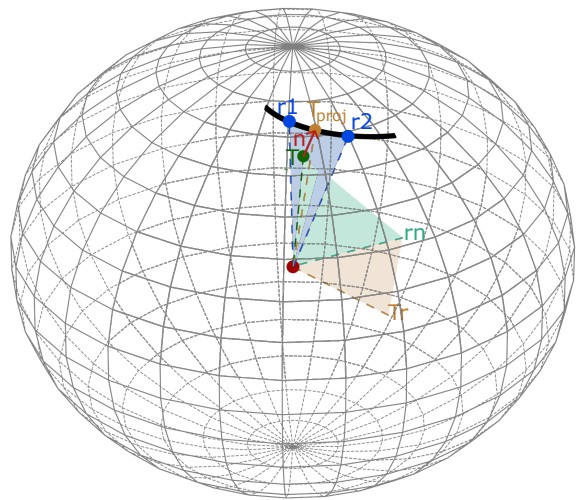

**Figure 6.** Illustration of the normal vector computation process using three consecutive cross products: r1 and r2 feature the two closest points to $T$ on the reference line, $\boldsymbol{rn}$ is the normal vector on the surface spanned by r1 and r2, $\boldsymbol{Tr}$ is the normal vector of $\boldsymbol{rn}$ with $T$, $T_{proj}$ is the projection point on the reference line and results by normalizing the cross-product of $\boldsymbol{rn}$ with $\boldsymbol{Tr}$ with the Earth's radius. $T$ and $T_{proj}$ then provide the normal vector $\boldsymbol{n}$.

neighboring cells of the closest points are used. Those are needed for the interpolation of the transports onto the reference line (as described in section 2.1.2). Again, any duplicate indices are removed automatically.

For the projection of the u and v velocities onto the strait, direction vectors and normal vectors for every grid-cell are determined using the functions `calc_dir_vec` and `calc_normvec`. Direction vectors are assumed to point from one grid-cell to the neighboring ones and are simply calculated by taking the difference between the Cartesian coordinates of $u_{x,y}$ and $u_{x+1,y}$ for $\boldsymbol{u}_{dir}$ and the difference between the Cartesian coordinates of $v_{x,y}$ and $v_{x,y+1}$ for $\boldsymbol{v}_{dir}$.

Normal vectors are calculated using three consecutive cross products. The basic principle is illustrated in Fig. 6. For each point $T$, $u$ and $v$, we find the two closest points r1 and r2 on the reference line. Transforming them into Cartesian coordinates, we can take their cross-product and get the vector $\boldsymbol{rn}$, standing orthogonally onto the surface spanned by r1 and r2 (blue surface). The cross-product of $\boldsymbol{rn}$ with $T$, yields the vector $\boldsymbol{Tr}$ (orthogonal to the green surface). Finally taking the cross-product of $\boldsymbol{rn}$ with $\boldsymbol{Tr}$ and normalizing the resulting vector with the Earth's radius, yields the point $T_{proj}$ on the reference line. The normal vector $\boldsymbol{n}$ for every grid-cell is given as the vector pointing from point $T$ to $T_{proj}$. The projection points $T_{proj}$ are later used as interpolation points on the reference line.

The function `proj_vec` uses the direct and normal vectors and equations 7 to calculate the projection vectors at every grid-cell. The actual calculation is done using `vel_projection`. Using the norm of the projection vectors, the $u$ and $v$ com-

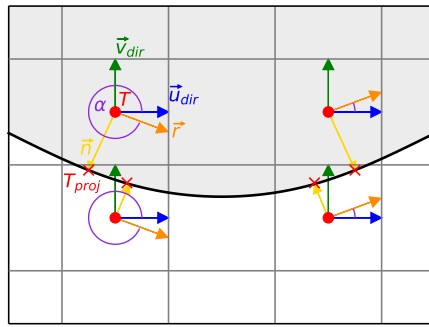

**Figure 7.** Determination of the $u$ component signs using the angle $\alpha$ between the direction vector of $u$ and the direction vector of the reference line r at the corresponding projection point. When $\alpha < 180$ the $u$ component is counted negative, when $\alpha > 180$ the $u$ component is counted positive. The v signs are calculated similarly using $\boldsymbol{v_{dir}}$.

ponents of every vertical layer are projected orthogonally onto the strait. To enhance conservation properties, we additionally multiply by the actual cell thicknesses of the cells before interpolation. The signs of the $u$ and $v$ components are determined by comparing the angles between direct and normal vector - the general idea is outlined in Fig. 7. Then, in a final step, the

scaled fluxes for every vertical layer are interpolated onto the reference line and divided by the respective layer thickness at the reference line.

The final result of `vel_projection` gives cross-sections of the velocities passing through the strait, with coordinates time, depth and x - the along strait distance. Exemplary cross-sections are shown in Sect. 3.1. To simplify the integration of the cross-sections to net transports, the horizontal (dx_int) and vertical extents (dz_int) of every point on the reference line are

output as well. Therefore, the net volume transport - similar to the end product of the LM - can be calculated by multiplying the section with dx_int and dz_int and summing up over x and depth.

Cross-sections of temperature and salinity are simply calculated by interpolating the scalar quantities of $T$ and $S$ onto the strait defined by the $T_{proj}$ points. To obtain the vertical profiles of heat and salinity transports, the $T$ and $S$ cross-sections have to be multiplied with the velocity cross-sections.

The velocity projection method has originally been developed for visualization purposes, however, as is shown in the next section, almost everywhere it provides nearly as accurate estimates of total fluxes through a strait as the LM.

## 3 Validation

In this section we will assess the robustness of the tools and their accuracy with multiple approaches. Firstly, the results of our computations are compared with naive calculation of the fluxes from interpolated velocity fields. Secondly, we specify simple $u,v$ and $T$ fields where the transports can be calculated analytically. These fields are then transformed exactly to the respective ocean model grids using the analytical mapping functions. The transports are then calculated using our LM and compared to the analytic solutions. Thirdly, we show the consistency between VPM and LM and then we check the correspondence between area integrated divergence fields and the transports through the array boundary. Lastly, we compare our results to results taken from an independent study where transports through Fram Strait were calculated by picking indices and signs for all grid points by hand. The exact definitions (start and end points) of all straits used throughout this paper are given in the Appendix (Tab. A1).

For verification we use harmonic functions to specify simple two dimensional $u,v$ and $T$ fields where the transports can be calculated analytically:

$$u(\lambda, \varphi) = 0; \qquad v(\lambda, \varphi) = v_r(\varphi) + v_0 \cos \varphi \sin k\lambda; \qquad T = T_r + T_0 \cos \varphi \sin(k(\lambda + \psi)) \tag{8}$$

with longitude $\lambda$, the latitude $\varphi$, wavenumber $k$ and phase shift $\psi$ given in radians. With the Earth radius $a$, we get for the transport:

$$F(\varphi) = a \cos \varphi \int_0^{2\pi} Tv \, d\lambda \quad = v_r(\varphi) T_r \cos \varphi + \frac{v_0 T_0 \cos^3 \varphi}{2} \cos \psi \tag{9}$$

The second term on the right-hand-side of (9) vanishes for phase shift $\pi/2$. The defined $v$ and $T$ fields are transformed to different ocean model grids (CMIP6) - the four used modelling grids are shown in Fig. 8 top and were chosen to be as different as possible in terms of horizontal resolution, number and location of poles, strength and extent of the distortion and used Arakawa partition. Transports are then calculated for full parallels at different latitudes ($\varphi$ = -70°N to 85°N) and for different wavenumbers (k = 1 to 100) using our LM, which leads to the generation of nontrivial polylines depending on the curvature of the respective grid. Solutions are then compared to the analytic solutions of the transport integral (eq. 9). Differences remain low (mostly below ±1%) for all four grid types over all assessed latitudes and wavenumbers as shown in Fig. 8 bottom. The biggest errors occur for the lower resolution grids at higher latitudes and higher wavenumbers. This is most likely caused by the coarse resolution and discretization of the models, which are not able to resolve the smaller generated waves, and less by the curvature of the grid alone, as the higher resolution model features very low errors up to $k$=100. At latitudes with regular grid lines errors due to discretization are deemed to be small. We calculated transports using the defined spherical harmonics for the tripolar CanESM5 grid at 20°N for various horizontal resolutions and found a difference of just about 0.015% between a 1° and a 10° resolution. Other small differences are mainly caused due to inaccuracies in the latitude selection. While analytic solutions are calculated at full latitudes, the position of the polylines may be shifted north or south due to the grids resolution.

For instance, the latitudinal shift of the 20°N line in the CanESM5 model leads to an error of 0.2%, explaining practically the total recorded error. Further, differences in the Earth radius (we assume $a = 6371$km) may lead to minor discrepancies.

The grey areas in the lower left panels of Fig. 8 indicate missing values and are due to the nature of the two dipolar grids, as they skip areas over the Greenlandic ice cap around the artificial North Poles. This is problematic for this application, as we define complete parallels for our analytic solutions. However, it does not affect actual transport calculations as this complication does not occur over oceanic areas.

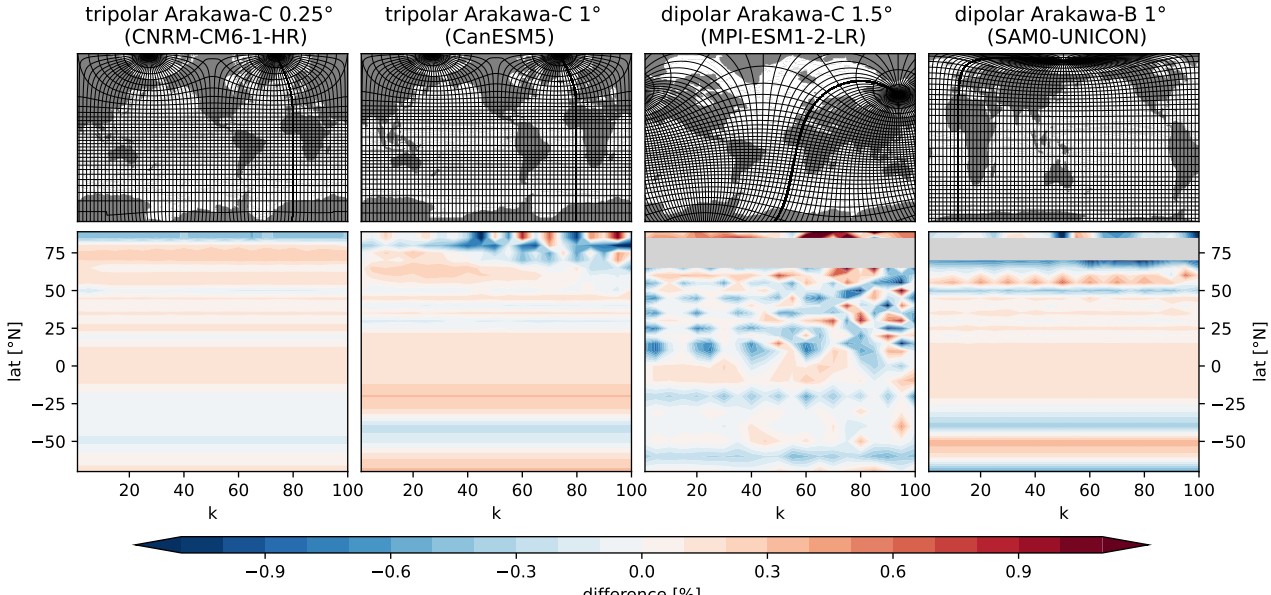

**Figure 8.** Top: grids used for the generation of polylines for the transport calculation via LM. The gridlines show the position of the regular gridlines on the distorted modelling grids (resolution of $5°$). Bottom: differences between the LM and analytic solutions of transports using spherical harmonics as fields. Grey areas in the bottom plots indicate areas where the calculation was not performed due to the absence of grid points in the dipolar grids over parts of Greenland.

In summary, results from the LM correspond very well with analytical results.

Interpolation of the vector components $u$ and $v$ onto regular grids is quite complex and may lead to significant errors in the calculated transports. The complexity arises from the rotation of the $u$ and $v$ components in comparison to the directions on a geographic latitude-longitude grid. Regridding would involve rotating the ocean velocity components to the new flow direction (eastward/northward) prior to the interpolation as done e.g. by He et al. (2019). However, for the rotation the exact grid angle at each grid cell is needed, which is not standard output for most CMIP6 models and reanalyses. Outten et al. (2018) found that small inaccuracies in the used angles, e.g. the exact position of the angles in the grid cell (center vs. cell edges or corners) may lead to differences in the calculated transports. Even if the model configurations and grid angles were archived correctly,

it is still hard to guarantee the conservation properties of the interpolated fluxes. Fig. 9 compares transports calculated from interpolated $u$ and $v$ values on a regular grid with those derived from $u$ and $v$ on the native grid for the CanESM5 model. Transports are calculated through Drake passage and the RAPID array, two straits where the native grid of the CanESM5 model is not distorted, and therefore any errors connected to the rotation of the velocity components are avoided. Even here

interpolation (both bilinear and conservative as defined in xESMF) leads to significant deviations from the actual transports obtained through both StraitFlux methods. Especially so at the RAPID array, a very long strait with a relatively small net volume transport. It is thus clear that the use of interpolated vector components is inappropriate for all kinds of transport calculations. An alternative approach would be to write each vector in terms of scalar vorticity and streamfunction using Helmholtz decomposition (e.g., Watterson, 2001), remap those scalar quantities to a regular grid and then recover eastward and

northward velocity components using gradients.

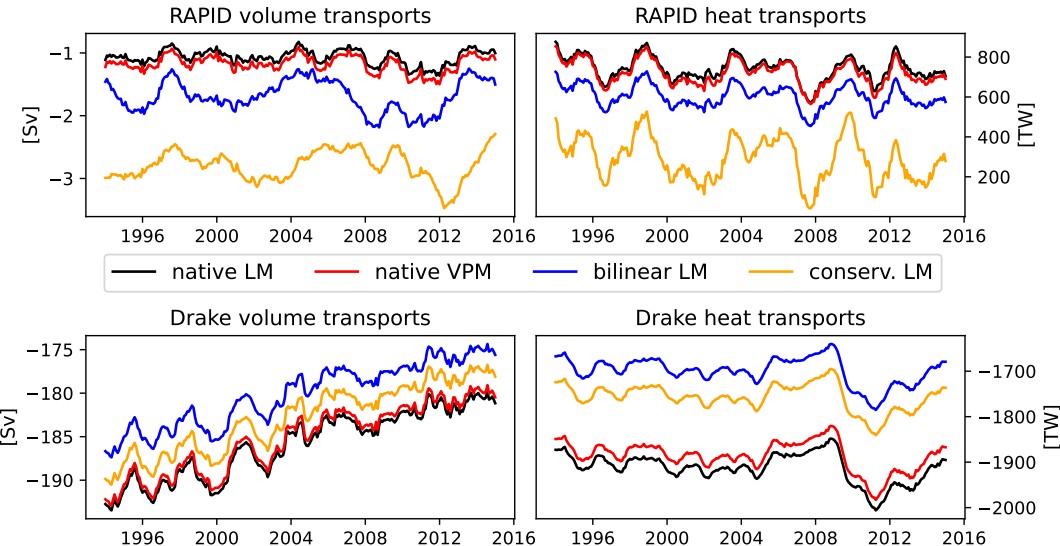

**Figure 9.** Display of the interpolation error for volume (left) and heat (right) transports at the RAPID array and at Drake passage from the CanESM5 model ($1°$ resolution).

While we have not compared the cross-section method with the analytical solution as we did for the LM, we show the credibility of the VPM by comparing volume, heat and ice transports obtained through the LM and the VPM (Fig. 10). Ideally, both methods should provide the same results, however due to differences in the calculation process small differences are expected. We choose a strait in the Arctic region - Fram strait - in order to come close to the strongest distortion of the curvlinear grids and

show volume transports for three models that use different grid types and Arakawa partitions. The CMCC-CM2-SR5 model uses a tripolar grid with an Arakawa-C partition, IPSL-CM6A-LR (Boucher et al., 2020) uses a tripolar grid with an Arakawa-B partition and SAM0-UNICON uses a displaced dipolar grid with an Arakawa-B partition. Depending on the model, transports

obtained through the different calculation methods match within a couple of percent of their total value. We consider this a very good result, given that this method was more designed for plotting purposes than for maximum accuracy of the integrated result.

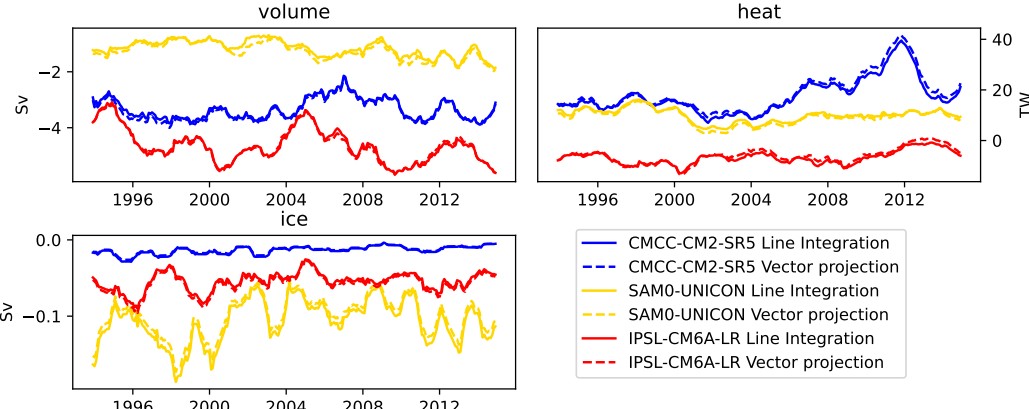

**Figure 10.** Comparison of volume, heat and ice transports obtained through the LM (solid) and through integration of cross-sections obtained through the VPM (dashed). The selected models use different grid types (see text) all with a horizontal resolution of about $1°$.

Another test to validate our transport calculation tools is the comparison of the transports across a whole latitudinal circle to the divergence of transports north of that latitude (i.e., validation of the sentence of Gauss). This is done for the ORAS5 ocean reanalysis (Zuo et al., 2019) and shown in Fig. 11. While the VPM differs from the values obtained by the LM and the divergence integral, the differences are still very small compared to those found in Fig. 9. Those differences may be caused by the increasing difference in integration area between the two methods with stronger grid curvature further north or also by an inaccuracy in the treatment of the North Fold boundary points. This needs to be further investigated and may be resolved in a later version of the software.

Lastly, we compared our methods to transports obtained by Heuzé et al. (2023) available via PANGAEA (Zanowski et al., 2023), who calculate transports of salinity, heat and volume through Fram strait for various CMIP6 models by choosing the coordinates for each model by hand. Fig. 12 shows the comparison of our transports to those obtained by Heuzé et al. (2023) for 10 selected models. For most models the results match within an expected range of uncertainty - differences may arise from differences in the exact positioning of the straits and differences in the definitions of $\rho$ and $c_p$.

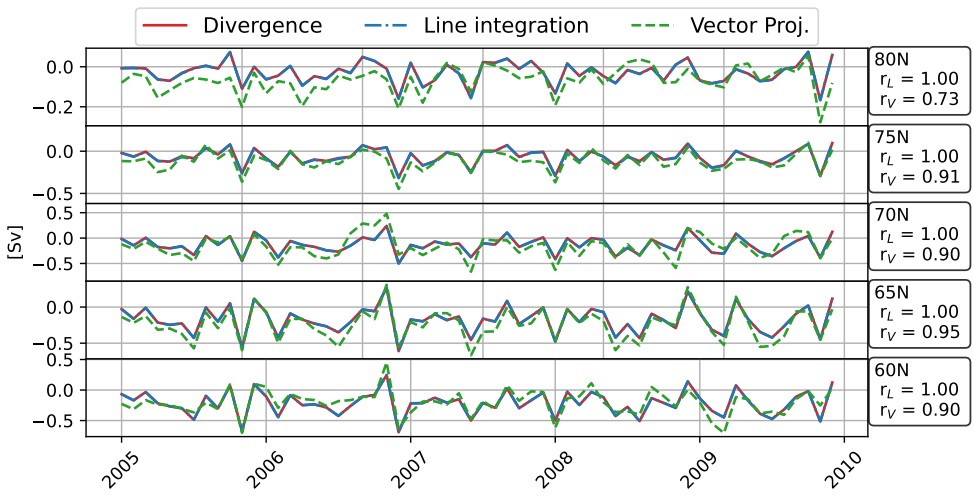

**Figure 11.** Integrated volume fluxes across different circles of latitude derived from ORAS5. Transports computed using the LM and VPM as well as through the integration of the divergence of transports north of the section in question.

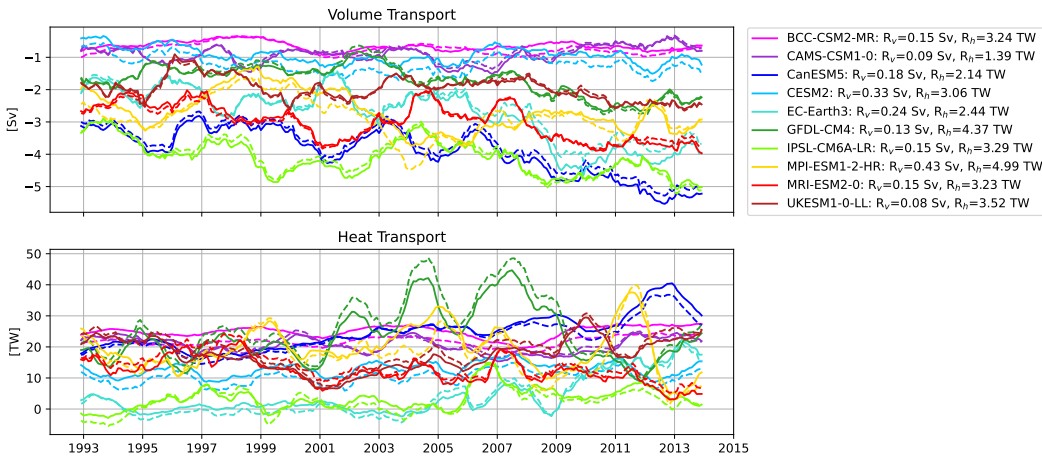

**Figure 12.** Volume and heat transport time series at Fram Strait for ten selected CMIP6 models from Heuzé et al. (2023) (solid) and our LM estimates (dashed). RMS-differences between Heuze's estimates and ours for volume ($R_v$) and heat ($R_h$) transports over the given time range are added in the line labels.

## 3.1 Application examples

To illustrate the abilities of StraitFlux we present some sample results and refer to studies where the tools have already been successfully used.

Results from the LM and VPM have already been shown in the validation section. Additionally to the net integrated transports the VPM also provides cross-sections of the vertical plane. Fig. 13 shows exemplary cross-sections of currents, temperature and salinity for the Greenland-Scotland Ridge (GSR) for two CMIP6 models with different horizontal resolutions - note the big difference in bottom topography and also the depiction of individual currents between the models.

Mayer et al. (2023b) use StraitFlux to compare oceanic transports across the GSR from ocean reanalyses against largely independent observations. They use the results from StraitFlux to partition the water masses into Atlantic, Overflow and Polar water, enabling a more in-depth analysis. They find that ocean reanalyses underestimate the observed Atlantic Water inflow by up to 15%, causing a low bias in oceanic heat transports (OHT) of 5%–22%. Further, they attribute a pronounced anomaly in OHT during the two-year period around 2018 to a reduction in Atlantic Water inflow through the Faroe–Shetland branch in combination with anomalously cool temperatures of Atlantic Water arriving at the GSR due to a recent strengthening of the North Atlantic subpolar gyre. Winkelbauer et al. (2024) use StraitFlux to calculate net transports of volume and heat passing into and out of the Arctic through Fram Strait, Davis Strait, Bering Strait and the BSO. They assess the transports' seasonal cycles and find clear correlations between oceanic transports and the Arctic's mean state. Fritz et al. (2023) use StraitFlux to assess transports in the Indonesian Throughflow (ITF) region and find reasonable agreements between reanalysis-based transports and observations in terms of means, seasonal cycles, and variability. Furthermore, transports have been calculated at the RAPID and OSNAP sections.

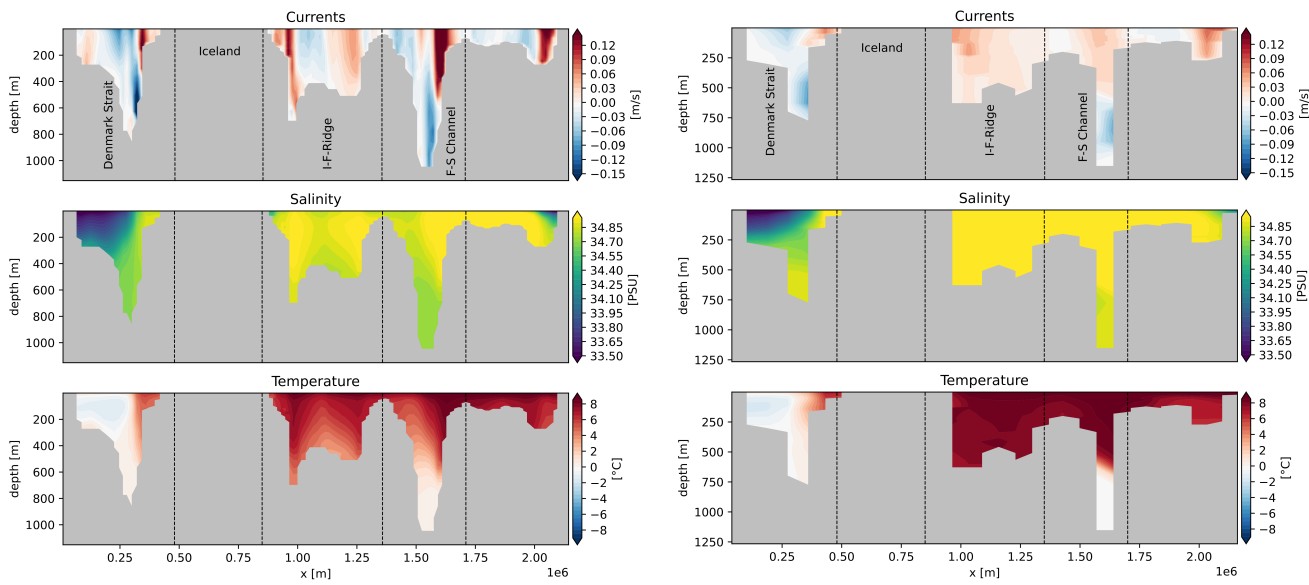

**Figure 13.** Sample Cross-sections of currents (top), salinity (middle) and temperature (bottom) for the Greenland Scotland Ridge for the CNRM-CM6-1-HR (0.25 deg horizontal resolution; Voldoire et al., 2019) and CNRM-CM6-1 (1 deg horizontal resolution; Voldoire et al., 2019) CMIP6 models.

## 3.2 Computational performance

The transport calculations usually need to involve only a small fraction of the 3D field values stored in the CMIP or reanalysis archives. As the current archives do not support extraction of subareas, the global fields need to be downloaded and consequently a fair amount of the total computational time is spent in reading and preprocessing the files.

In order to calculate e.g. the temperature flux, the following fields from a typical 1/4 degree CMIP6 model are needed:

| Field | approx. Dimensions | approx. Size [MB] |
|---|---|---|
| Sea Water X Velocity $u$ | 1400x1000x75 | 300 |
| Sea Water Y Velocity $v$ | 1400x1000x75 | 300 |
| Sea Water Potential Temperature $T$ | 1400x1000x75 | 300 |
| Ocean Model Cell Thickness thkcello | 1400x1000x75 | 300 |

**Table 1.** Approximate dimensions and sizes of variables for typical 1/4 degree models needed for the transport calculations of one time step.

To detect the indices of the section, calculate horizontal meshes and determine the Arakawa partition the software reads one vertical layer of the data files. Once the section is known, the software chooses the subregion, so that not the whole files need to be read. While reading the files takes up the majority of the calculation time, the calculation itself is performed relatively fast. Approximate times for the major calculation steps for a Xeon Gold 6148 CPU for a 1 degree model (CanESM5) and a 1/4 degree model (EC-Earth3P-HR) at Fram Strait are given in the table below. The calculation of the mesh files and detection of the Arakawa partition has to be performed only once per model, and the calculation of indices and parameters for the VPM (normal and direct vectors, signs of velocity components etc.) once per model and strait, when the functions' parameter `saving` is set to `True` (default). This speeds up subsequent calculations, e.g. for different months or straits, considerably.

With monthly time resolution, for 1/4 degree models it is possible to calculate transports directly for multiple years (e.g. the calculation of the 65 year period takes about 60 s), for higher resolution models we advise to loop the calculation (e.g. over 12 months) to avoid high memory consumption. For faster performance calculations may of course be done in parallel. Also the flux calculation for other ensemble members can be done in parallel as well.

## 3.3 Availability

StraitFlux is available as open source python package at github and zenodo and can be installed from pypi. The github repository also contains an example script and some example datasets as well as a requirements file, to simplify the installation and usage of StraitFlux.

| horizontal resolution [°] | 1°x1°, 75 l, 12 m | 0.25°x0.25°, 75 l, 12 m |
|---|---|---|
| read files for mesh and index calculation ** | 1.5 s | 3.5 s |
| calculate indices ** | 0.9 s | 1.4 s |
| determine Arakawa partition * | 3 ms | 3 ms |
| read files (subselected) | 2.2 s | 3.5 s |
| calculate dz at cell faces ** | 25 ms | 0.15 s |
| Line Integration Method (LM) | | |
| calculate mesh files * | 10 s | 44 s |
| calculate transports | 50 ms | 80 ms |
| **Total** | 15 s / 2.3 s | 55 s / 3.8 s |
| | | |
| Vector Projection Method (VPM) | | |
| calculate projection vectors and constants ** | 8.2 s | 31 s |
| calculate transports | 0.2 s | 3.3 s |
| regrid to section | 0.1 s | 0.2 s |
| **Total** | 12.5 s / 2.5 s | 43 s / 7.5 s |

**Table 2.** Approximate calculation times for 12 months of monthly data for two exemplary models with 1 degree and 1/4 degree resolution and 75 vertical layers at Fram Strait. Functions are divided into "used by both" (top), LM (middle) and VPM (bottom). The asterix * indicates functions that only have to be calculated once per model and ** functions that need to be calculated once per model and strait. Total times are given for the first calculation performed (left values) and every consecutive calculation (right values), when the functions' parameter 'saving' is set to 'True' (default).

StraitFlux is a free software and can be redistributed and/or modified under the terms of the GNU General Public License version 3 as published by the Free Software Foundation.

## 4   Conclusions

In this study, we have introduced StraitFlux, an open-source Python package designed to facilitate the calculation and analysis of oceanic transports through arbitrary oceanic straits and sections. We give a comprehensive overview of StraitFlux, including its underlying principles, software implementation, validation, and application examples. StraitFlux facilitates scientific studies to validate models and gain valuable insights into ocean circulation, heat transports, and water mass exchanges, making it a useful tool for climate scientists, oceanographers and modelers.

StraitFlux works on various curvilinear ocean modeling grids and is written so flexible that it is expected to work for future versions (e.g. CMIP7) as well. Unstructured grids are not included in this release. However, the methods have already been successfully adapted and tested for the FESOM2 ocean model (Danilov et al., 2017), the successor of FESOM which is for instance used in the AWI Climate Model (AWI-CM) in CMIP6, and are planned to be included in future versions of StraitFlux

as well. The tools include two methods for calculating oceanic transports: the Line Integration Method (LM) and the Vector Projection Method (VPM). The LM creates a closed polyline along grid cell faces to compute net integrated transports, while the second method employs vector projection algorithms to estimate the share of u and v components passing orthogonally through the strait and generates cross-sections of velocities, temperatures and salinities in the vertical plane.

Both methods have been thoroughly validated and produce reliable results across various ocean models and grids. Our validation efforts have demonstrated that StraitFlux consistently matches analytical solutions, even in complex grid configurations and regions with strong distortion. Both methods deliver net transports that match within a couple % of their total value, even at the most distorted sections. The tool's accuracy is further affirmed by comparisons with the divergence of transports and independent transport calculations. One problem remains at the northernmost latitudes for the VPM, which we hope to resolve soon.

The applications of StraitFlux extend to a wide range of research areas. Researchers can use the package to analyze seasonal cycles, mean states, and variability in oceanic transports. Furthermore, the ability to generate cross-sections of currents, temperature, and salinity provides a detailed view of the ocean's vertical structure and flow patterns.

In summary, the user-friendly implementation and broad applicability make it a valuable tool for studying the Earth's climate system and its dynamics. The simplified comparison to observational data highlights its suitability for model validation and assessment. We hope that StraitFlux empowers researchers to explore and understand oceanic transports more thoroughly, given their importance in the climate system and changes therein.

*Code availability.* The Python implementation of StraitFlux is available at https://github.com/susannawinkelbauer/StraitFlux (last access: 19 March 2024) and can be installed from pypi. The github repository additionally contains the notebook 'Examples.ipynb' with some easy examples to get started with the transport calculations. Data files used in the notebook may be downloaded via ESGF (https://esgf-node.llnl.gov/search/cmip6/). Version 1.0.4 of StraitFlux, which is described and used in the paper, is long-term archived at zenodo (Winkelbauer, 2024).

*Data availability.* CMIP6 data used in the validation section of this paper may be downloaded via ESGF (https://esgf-node.llnl.gov/search/cmip6/). Reanalyses data is available via the Copernicus Marine Service (https://marine.copernicus.eu/).

| Strait | start point (lat[°N],lon[°E]) | intermediate points (lat[°N],lon[°E]) | end point (lat[°N],lon[°E]) |
|---|---|---|---|
| Fram | (78.82,-20.7) | - | (78.83,12.00) |
| Davis | (66.65,-61.80) | - | (67.31,-52.50) |
| Barents Sea Opening | (78.00,18.00) | - | (69.20,19.80) |
| Bering | (65.99,-170.50) | - | (66.75,-166.00) |
| Greenland Scotland Ridge | (68.53,-30.82) | (66.01,-23.24), (64.41,-15.07) (62.07,-6.87), (60.28,-1.17) | (59.47,6.11) |
| RAPID | (26.00,-80.50) | - | (26.00,-13.50) |
| Drake | (-55.70,-66.92) | - | (-64.10,-59.20) |

**Table A1.** Start- and endpoints for the straits used. The net Arctic transports are calculated using the sum of Fram, Davis, Bering and Barents Sea Opening. The Greenland Scotland Ridge is defined as kinked line with 4 intermediate points.

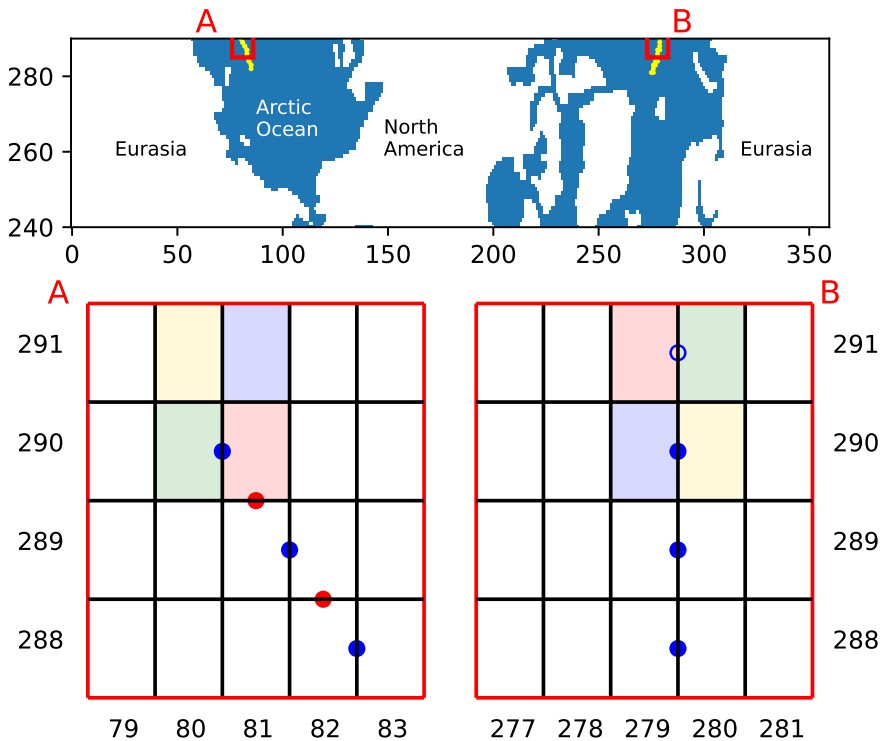

**Figure A1.** Indices selection across the northern boundary for the CMCC-CM2-SR5 model. The top two rows of grid cells are rotated along the northern boundary, colorful cells show duplicate cells which are pivoted at the top boundary (= same cells but upside down). Filled blue dots show selected u indices, filled red dots show selected v indices. Empty blue dot shows overlapping point which is not selected.

*Author contributions.* SW, MM and LH conceptualized the study. SW developed the Python implementation of StraitFlux with the help of MM and LH. SW performed the data analysis, including the production of the figures in the paper, and prepared the manuscript. All authors wrote, edited and reviewed the manuscript and agreed to the publication of the present version of the manuscript.

*Competing interests.* The authors declare that they have no competing interests.

*Disclaimer.* TEXT

*Acknowledgements.* Susanna Winkelbauer and Michael Mayer were supported by the Austrian Science Fund project P33177 and the Copernicus Marine Service contract 21003-COP-GLORAN Lot 7. The authors sincerely thank Céline Heuzé from the University of Gothenburg for the provision of manually calculated transport data to use as validation and helpful discussions on the transport calculation methodology. We acknowledge work by Vanessa Seitner who developed an earlier version of Straitflux. We thank Shizhu Wang and a second anonymous referee for their constructive comments that improved this study.

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
