# Peer review of "StraitFlux - Precise computations of Water Strait fluxes on various Modelling Grids"

_EGUsphere, 2023_

## Author Comment (AC2)

Thank you very much for your positive comments and constructive feedback, you addressed some important points. Your clarifications helped to make the manuscript clearer for the reader. Our responses are provided in green (changes made in the manuscript are written in **bold**) together with your original comments in black. We really appreciate your time and insight in reviewing our manuscript!

Kind regards,
Susanna (on behalf of all co-authors)

This paper presents the python package StraitFlux, and demonstrates how it can be used to calculate ocean strait transports for various ocean model grids. It seems like a valuable and easy to use tool, and I appreciate the effort put into making this a public tool that can save other users a lot of time. I recommend that this manuscript is published after some clarifications are made:

L4-5: A bit unclear what you mean by this sentence. Are you referring to the complications from the singularity at the true North Pole? Or errors in fields that are interpolated to standard lat/lon grid?
We refer to errors being generated through regridding of the velocity fields onto regular lat/lon grids. We clarified it to the following:
*Use of data **regridded to** standard latitude/longitude grids is not an option since transports computed from those are not mass **consistent and may feature substantial errors.***

L9: What exactly do you mean by the integrand for the second method? This word is not used for anything elsewhere in the manuscript.
We refer to the velocity/temperature/salinity cross-sections, which need to be integrated over depth and along the strait to yield the net volume/heat/salinity transports. We changed it in the manuscript:
*Apart from the input data on the original model grids the user only needs to specify the start and end points of the required section to get the integrated net transports **(for the first method) and cross-sections (for the second method). Integration of the cross-sections along their depth and horizontal extent also yields net transports.***

Eq. (4) ice is mentioned here, but not any further. Would be good to discuss further, e.g. by including an example of ice transport in the paper if mentioning it here.
We adapted Fig. 10 to also show heat and ice transports. And adapted the associated sentence in the manuscript (L301-302):
*While we have not compared the cross-section method with the analytical solution as we did for the LM, we show the credibility of the VPM by comparing **volume, heat and ice transports** obtained through the LM and the VPM (Fig. 10).*

[Figure]

*Figure 10. Comparison of volume**, heat and ice transports** obtained through the LM (solid) and through integration of cross-sections obtained through the VPM (dashed). The selected models use different grid types (see text) all with a horizontal resolution of about 1°*

L86: Maybe briefly explain why closed volume transport is not generally the case?

It is generally not the case for partial sections as transports are e.g. compensated by flows through other passages. Good examples are given by the four Arctic passages (Bering Strait, Fram Strait, the Canadian Archipelago and the Barents Sea Opening). Looking at the net flow into the Arctic, volume transports are balanced, however none of the individual straits feature balanced volume transports. This would generally introduce the dependence of the heat transports for the individual straits on the chosen temperature scale via T_ref.

We adapted the sentence as follows:

***Previous studies (Schauer and Losch, 2019; Schauer and Beszczynska-Möller, 2009)** correctly point out that true heat transports would actually demand closed volume transports through the examined straits. **This is generally not the case for partial sections as transports may be compensated by flows through other passages and unbalanced volume transports would generally introduce the dependence of heat transports on the chosen temperature scale via T_ref.***

Schauer, U. and Beszczynska-Möller, A.: Problems with estimation and interpretation of oceanic heat transport – conceptual remarks for the case of Fram Strait in the Arctic Ocean, Ocean Sci., 5, 487–494, https://doi.org/10.5194/os-5-487-2009, 2009.

Vector Projection Method: Is it using the same method for Arakawa A, B and C grids?

Yes, in general the Vector Projection Method is independent of the Arakawa partition. Only the cell thicknesses have to be transformed to the correct u/v positions on the grid cell.

Fig 2b. One of the cells is not in contact with the red line. From the text I understand that only cells in contact with the line is used.

We added a cell not touching the reference line, as for the Vector Projection Method

the cells in contact with the lines and also their neighboring cells are needed for the bilinear interpolation onto the reference line.

We clarified it in the text at L116-118:
*For every grid cell touching the strait **and their neighboring cells (needed later on for the interpolation onto the strait)** we calculate direction vectors of the u and v components (blue and green arrows), and normal vectors pointing from the tracer grid cell in the direction of the strait (yellow arrows).*

And at L123-125:
*The projection vectors' magnitudes are then used to compute orthogonal transports at all grid cells touching the strait **and their neighboring cells**. In the final step, these transports are interpolated bilinearly onto the closest points on the reference line (black crosses in Fig. 2b, called T_proj henceforth) and divided by the respective cell thicknesses on the reference line to obtain velocities.*

L124-126: Why do you convert velocities to transports and then after interpolation divide by the respective cell thicknesses to obtain velocities again, instead of just interpolating the velocities?
As pointed out at L47-49 and at L.292-295 and Fig. 9 interpolation of the velocities (=vector components) prior to the transport calculation may compromise the conservation properties of the model and therefore it may lead to quite substantial interpolation errors (see Figure 9) and unrealistic behavior in the simulated system, especially in regions with complex topography, sharp gradients or complex flow patterns. By calculating the orthogonal transports first (= scalar quantities) we ensure that the conservation properties of the models are maintained.

We changed the paragraph as follows:
*In the final step, these transports are interpolated bilinearly onto the closest points on the reference line (black crosses in Fig. 2b, called T_proj henceforth) and divided by the respective cell thicknesses on the reference line to obtain velocities. This results in velocity cross-sections of the vertical plane which are spaced irregularly along the along-strait distance (x) in accordance with the distribution of T_proj points. The interpolation onto evenly distributed points on the section, to e.g. enable the calculation of differences with other models/reanalyses, is initially left to the user and eventually will be included in a future version of StraitFlux. **By calculating transports (scalar quantities) prior to the interpolation onto the strait we ensure that the conservation properties of the models are maintained. This ensures that integration of the cross-sections** along the along-strait distance (x) and depth (z) provides net transports, which agree very well with the values obtained by the LM (see Sect. 3).*

L155: How is a straight reference line drawn between the endpoints? Is it following the shortest/great-circle distance along a sphere? Or can it be defined to e.g. follow a specific latitude?
When two endpoints are given the reference line is drawn following the shortest distance along the sphere. Therefore, if the 2 endpoints have the same latitude, the

strait will automatically follow this specific latitude. There is also the option to provide a list of individual coordinates by setting the functions 'set_latlon' to True and providing latitude and longitude points via the 'lat_p' and 'lon_p' parameters. We clarified it in the manuscript:

*The determination of section positions for transport calculations is accomplished in the def_indices function.* ***Users can specify the start and end points of a section using the 'coords' parameter in the 'transports' function, the section will then follow the shortest distance along the sphere. Alternatively, users may pass specific coordinates by setting the 'set_latlon=True' parameter and providing a list of latitude ('lat_p') and longitude ('lon_p') points. The latter option also allows the calculation of "kinked" sections.***

Write also somewhere how the example straits used in the paper are defined, e.g. specify the start and endpoints used. Or a latitude for the Fram Strait.
We added the coordinates (start and endpoints) as table in the appendix:

| Strait | start point (lat[°N],lon[°E]) | intermediate points (lat[°N],lon[°E]) | end point (lat[°N],lon[°E]) |
|---|---|---|---|
| Fram | (78.82,-20.7) | - | (78.83,12.00) |
| Davis | (66.65,-61.80) | - | (67.31,-52.50) |
| Barents Sea Opening | (78.00,18.00) | - | (69.20,19.80) |
| Bering | (65.99,-170.50) | - | (66.75,-166.00) |
| Greenland Scotland Ridge | (68.53,-30.82) | (66.01,-23.24), (64.41,-15.07) (62.07,-6.87), (60.28,-1.17) | (59.47,6.11) |
| RAPID | (26.00,-80.50) | - | (26.00,-13.50) |
| Drake | (-55.70,-66.92) | - | (-64.10,-59.20) |

**Table A1.** Start- and endpoints for the straits used. The net Arctic transports are calculated using the sum of Fram, Davis, Bering and Barents Sea Opening. The Greenland Scotland Ridge is defined as kinked line with 4 intermediate points.

And added the following sentence in the Validation section:
***The exact definitions (start and end points) of all straits used throughout this paper are given in the Appendix (Tab. A1).***

L167: How are the points i on the refline defined?
i is the index that we use to select the points along the equally spaced reference line. In the newest version of StraitFlux spacing of the reference line is dependent of the resolution of the used modelling grid (see Reviewer 2 minor comment 7). E.g. 0.1° spacing for a model with 0.25° horizontal resolution, would mean that i moves along the reference line with 0.1° steps.

Following a comment from Reviewer 2 we changed Fig.3 to be easier to comprehend and changed the caption to the following:

[Figure]

Figure 3. Illustration of the indices selection process using select_points. **Lines of constant latitude/longitude are shown in grey.** a) and b) Determination of consecutive grid-cells **on the native grid** by comparing distances d (orange lines) of 4 surrounding grid-cells **for all equally spaced points i along the reference line.** c) Specification whether a u or v component should be taken.

We adapted the indexing for Fig3c) in the text (L176-178):
For instance, to get to cell **j** in Fig. 3c, we came from left, hence the v component of cell **j** is taken. In order to get to cell **j** + 1 we come from below, therefore the u component of cell **j** + 1 is taken.

And we added the following footnote:
**Note that we follow the native grid points (x,y) and the local direction of x is not necessarily west-east and the local direction of y not south-north. For instance, coming from left means coming from point [$x_{i-1}, y_i$] to point [$x_i, y_i$].**

L230-231: I don't understand why the difference between velocity components is taken? Has it something to do with correcting the positions on the Arakawa grid?

To calculate the projections for the vector projection method we need vectors pointing in the direction of the velocity components, we call them "direct vectors" (= u_dir and v_dir in Eq.7; blue and green arrows in Fig. 2b) pointing from one u/v-point to the next u/v-point. To obtain the direct vectors we simply take the difference between the u/v points (in cartesian coordinates) of two neighboring cells which gives us the desired vector between those points.

Possibly we created some confusion using the term direct vectors instead of direction vectors, so we changed it to the following:
For the projection of the u and v velocities onto the strait, **direction** vectors and normal vectors for every grid-cell are determined using the functions calc_dir_vec and calc_normvec. **Direction** vectors are assumed to point from one grid-cell to the neighboring ones and are simply calculated by taking the difference between the **Cartesian coordinates of** $u_{x,y}$ and $u_{x+1,y}$ for $u_{dir}$ and the difference between the **Cartesian coordinates of** $v_{x,y}$ and $v_{x,y+1}$ for $v_{dir}$.

Fig 7: I cannot fully understand this figure, and how the angle alpha is defined. Intuitively, I would think u was defined negative if the absolute value of the angle between r and u_dir was more than 90 degrees.

Alpha is the angle between u_dir and the strait (=vector r). If the angle is smaller than 180 degrees u is defined negative, and if it's >180° u is defined positive. **It's not about u pointing to the east or west, we always consider the positioning of u to the strait.** In Fig. 7 transports into the grey area are defined positive and out of it negative. Due to the curvature of the strait the 2 points on the left pass the strait (black bold line) from the other side than the points on the right side. The angle between u_dir at the upper left and lower left points and r at those points is >180° and therefore u_dir is counted positive. However, for the two points on the right, the angle is <180° and u_dir is counted negative.

We adapted the figure and included the r-vectors and the alpha-angles to make it easier to comprehend:

[Figure]

L241-243: I don't understand this sentence, and why the u and v components are scaled.
The formulation was a bit misleading. By "scale" we actually meant the projection of u/v onto the orthogonal. With "scale" we just meant to express that u/v have to be multiplied by the norm of the projection vector as not 100% of u/v actually pass through the strait. [for flows parallel to the strait actually 0% pass through ("scaling" factor of 0) and for flows orthogonally onto the strait 100% pass through ("scaling" factor of 1)]
We changed the formulation to omit the phrase "scale":
*Using the norm of the projection vectors, the u and v components of every vertical layer* **are projected orthogonally onto the strait. To obtain cross-sections of velocities we additionally** *multiply by the actual cell thicknesses of the cells.*

L293: What is ESMFgrid? Do you mean xesmf's Regridder?
Yes, we changed it to:
*We compare both, bilinear and conservative interpolation as defined* **in xESMF, a python package for regridding,** *and calculate volume and heat transports through the sum of*

*Arctic main gateways (Fram Strait, Barents Sea Opening, Davis Strait and Bering Strait) for the CanESM5 model.*

Fig 12. Maybe add in figure title or text that this is the Transports for the Fram Strait. We added it to the figure caption:
*Figure 12. Volume and heat transport time series **at Fram strait** for ten selected CMIP6 models from Heuzé et al. (2023) (solid) and our LM estimates (dashed).*

L339: ITF: please define abbreviation
We defined it as Indonesian Throughflow.

If possible, can you sketch an example showing why interpolation methods make such large errors? Or cite eventual other papers who discuss why? I find it a little hard to imagine how interpolation errors can give such large errors in the net Arctic volume transport.
The net Arctic volume transport is quite sensitive to small inaccuracies in the calculation, as large transports through the individual straits (e.g. >2Sv for the Barents Sea Opening) add up to a very small net transport (about 0.2Sv). Nevertheless, we discovered an error in our interpolation from curvilinear to native grid concerning the rotation of the vector components. Therefore, we repeated the calculation for Drake Passage and the RAPID array (see Figure below), two sections where the grid of the considered CanESM5 model is regular and therefore the rotation of the vector components may be omitted. Interpolation errors are smaller than what we showed in Fig. 9, however especially for the RAPID array, a very long strait with a relatively small net volume transport, they are still quite significant. Complications due to the rotation of u/v will possibly introduce even larger errors further north (although probably smaller than what we've assumed before). We changed the figure and revised the whole paragraph:
***Interpolation of the vector components u and v onto regular grids is quite complex and may lead to significant errors in the calculated transports. The complexity arises from the rotation of the u and v components in comparison to the directions on a geographic latitude-longitude grid. Regridding would involve rotating the wind components to the new wind direction (eastward/northward) prior to the interpolation as done e.g. by He et al., 2019 (https://doi.org/10.1038/s41558-018-0387-3). However, for the rotation the exact grid angle at each grid cell is needed, which is not standard output for most CMIP6 models and reanalyses. Outten 2018 (https://doi.org/10.1175/JreCLI-D-18-0058.1) found that small inaccuracies in the used angles, e.g. the exact position of the angles in the grid cell (center vs. cell edges or corners) may lead to differences in the calculated transports. Even if the model configurations and grid angles were archived correctly, it is still hard to guarantee the conservation properties of the interpolated fluxes. Fig. 9 compares transports calculated from interpolated u and v values on a regular grid with those derived from u and v on the native grid for the CanESM5 model. Transports are calculated through Drake passage and the RAPID array, two straits where the native grid of the CanESM5 model is not distorted, and therefore any errors***

*connected to the rotation of the velocity components are avoided. Even here interpolation (both bilinear and conservative as defined in xESMF) leads to significant deviations from the actual transports obtained through both StraitFlux methods. Especially so at the RAPID array, a very long strait with a relatively small net volume transport.*
*An alternative approach would be to write each vector in terms of scalar vorticity and streamfunction using helmholtz decomposition, remap those scalar quantities to a regular grid and then recover eastward and northward velocity components using gradients.*

[Figure]

*Figure 9. Display of the interpolation error for volume (left) and heat (right) transports at the RAPID array and at Drake passage from the CanESM5 model (1° resolution).*

Typos:

L268: radiant - should be radians
We corrected it.

L313: VPN – should be VPM
We corrected it.

L360: Gitlab – maybe you mean github instead, since the urls provided go to github?
Yes, we changed it to github.

L373 i.a. – should be i.e. (?)
We changes it to "for instance".

L383: northern most – cut space between words
We corrected it.

L397: downlaoded – should be downloaded
We corrected it.

---

## Author Comment (AC3)

Thank you very much for your positive comments and constructive feedback, you addressed some important points. Your clarifications helped to make the manuscript clearer for the reader. Our responses are provided in green (changes made in the manuscript are written in **bold**) together with your original comments in black. We really appreciate your time and insight in reviewing our manuscript!

Kind regards,
Susanna (on behalf of all co-authors)

Given the important roles of ocean circulation in shaping the climate system, correctly diagnosing the volume/salinity/heat transport in the ocean models is always important but complicated by different model grid configurations. Winkelbauer et al. present two new methods in calculating ocean transport, with one sticking to the original model grid lines and the other following strict vector projection and interpolation. Both methods work well with different Arakawa grids, and the results given by these two methods are very similar to assumed analytical results. And I would say the code is versatile in that users can define strait sections in different ways (to my understanding, 3 ways in defining the straits). What makes me more satisfied is that the code is organized into a Python package and maintained in GitHub, which facilitates its future upgrade and public use. Both the paper and the code are well written and I think the paper should be published after considering/answering my following suggestions/questions.

**Major concerns:**

I know that this is a scientific rather than a technical documentation, but some technical details are still needed to help the readers understand the calculation process.

1. Line123-125: I have difficulty in understanding the processes given here. Are the **transports or projection vectors** "interpolated bilinearly onto the closest points on the reference line? Put it another way, after the projection vectors of all touched cells are calculated, what is the next? Do we (1) calculate transports using these projection vectors of the touched cells (which might need the length of the reference line that falls into each grid cell), then interpolate the transport value onto the closest points on the reference line, or (2) interpolate the projection vectors onto the reference line. Also, the transports are "divided by the respective cell thicknesses on the reference line". What do you mean "thickness" here? Vertical thickness or the length of the horizontally overlapped part? I suggest the authors to rewrite these 3-4 sentences and give it in a more clear way.

   (1) is the correct assumption, however the term "transport" was probably a bit misleading. We use the projection vectors and multiply the projected u and v components with the respective vertical thickness (different before and after the interpolation), but not with the respective length of the reference line (not

needed as we would need to divide by exactly the same length after the interpolation to obtain velocities again). This is done for every grid cell that touches the reference line and it's neighboring grid cells (those are needed for the interpolation in the next step, see minor comment 5), the "transports" are then interpolated onto the closest points on the reference line (T_proj) and to obtain velocities again we divide by the vertical thicknesses/extents at the T_proj points.

We changed the sentences to the following:
***Using the magnitudes of the projection vectors we calculate the u and v components pointing orthogonally onto the strait at all grid cells touching the strait and their neighboring cells (needed for the bilinear interpolation). Then, we multiply them with the respective vertical cell thicknesses at the u/v points and interpolate those orthogonal "transports" bilinearly onto the closest points on the reference line (black crosses in Fig. 2b, called T_proj henceforth). In a final step we divide by the vertical thickness of the cells on the reference line to obtain velocities again.***

2. **Line190**: Section2.2.3 talks about the halo grids (some people call these halos or halo grids. E.g., https://github.com/pangeo-data/xESMF/issues/109). This section deserves more sentences because the authors only said "these conditions have to be handled with care" but how? A detailed discussion on the halos might go beyond the scope of the current manuscript, but I would like to see more discussion on how to overcome the halos in order to a smooth use of StraitFlux. An example on dealing with the halo grid problem would be even better.

   In the newest version of StraitFlux we implemented an automatic check where the algorithm checks whether any overlapping points exist at the cyclic or the northern boundary points, and if so, those are removed. While quite strait forward with the cyclic points, it's a bit trickier to account for all the different handlings of the northern boundary points. We tested the indices selection for a strait going "over" the northern boundary of the tripolar grids for multiple models with different boundary definitions (CanESM5, CMCC-CM2-SR5, ACCESS-CM2, CAMS-CSM1-0, IPSL-CM6A-LR, EC-Earth3) and in all cases the new version of StraitFlux selected the correct indices to avoid duplicates and/or gaps. Below a figure of the indices selection for the CMCC-CM2-SR5 model, which pivots the 2 top points at the northern boundary. The colorful cells show "duplicate points which are pivoted" onto the "other" side (see jump in x indices). The blue and red filled dots show the selected u and v indices. The blue "empty" dot in the right panel corresponds to the top blue filled dot in the left panel and is **not** used. Even for a complicated case as this no cells are counted twice and also no gaps are present (we have a u index between the green and red cell (left, index 80) and between the neighboring blue and yellow cell (right, index 279)). While the selection worked perfectly fine for the tested models, we still can't guarantee that it will work for all

possible conditions, therefore the code still outputs the warning: "Attention: Strait crossing the northern boundary – make sure correct indices are chosen!". We added the figure to the appendix and adapted the text as follows:

[Figure]

*Figure A1. Indices selection across the northern boundary for the CMCC-CM2-SR5 model. Colorful cells show duplicate cells which are pivoted at the top boundary. Filled blue dots show selected u indices, filled red dots show selected v indices. Empty blue dot shows overlapping point which is not selected.*

These conditions have to be handled with care, as especially the volume transport calculation is very sensitive and can yield useless results when there is a gap in the integration line or if any grid-cells are counted twice. **StraitFlux automatically checks for overlapping cyclic boundary points and drops any duplicates, this should ensure correct transport calculations across the zonal boundaries independent of how the models deal with periodicity. Similarly concerning the north boundary conditions StraitFlux should automatically select the correct indices and avoid gaps and/or duplicates. We tested this successfully for an arbitrary line going over the top boundary of the model grids for various CMIP6 models with different boundary conditions (CMCC-CM2-Sr5, EC-Earth3, CanESM5, ACCESS-CM2, CAMS-CSM1-0, IPSL-CM6A-LR). Fig. A1 in the appendix shows an example for the CMCC-CM2-SR5 model. The top two rows of grid cells are rotated along the northern boundary, colorful cells show duplicate cells which are pivoted at the top boundary (= same cells but upside down). StraitFlux correctly chooses the indices so that a continuous line without overlaps is formed. While the indices selection worked for the tested models, the generated indices should still be checked to ensure a**

***continuous line also for more complicated boundary conditions. Therefore, the code automatically outputs the warning "Attention: Strait crossing the northern boundary – make sure correct indices are chosen!" when moving across the boundary of the grid.***

**Minor concerns:**

Dear

1. Line5: transports computed from those are not mass/**volume**
   *We changed it to:*
   *Use of data **interpolated to** standard latitude/longitude grids is not an option since transports computed from **interpolated velocities** are not mass consistent.*

2. Line45: it's not just the artificial meridional velocity. The artificial zonal velocity does not point to the true east either.
   *we adapted the sentence to the following:*
   *While solving the numerical problem of a singularity over the ocean, those curvilinear grids complicate the calculation of oceanic transports, especially in the proximity of the poles, as velocities in the direction of the artificial poles do not point in the direction of the true north **and artificial zonal velocities do not point to the true east**.*

3. Line46: "angle of the grid-lines" is always 90 degree since we are using general orthogonal curvilinear coordinates. I think you mean the angle between gridlines and regular lon-lat lines.
   *Yes, we adapted it:*
   *The exact position of the poles, **the angle between the native gridlines and regular longitude-latitude lines, as well as the horizontal and vertical resolution** varies between different models, forming a vast amount of different grid types that complicate inter-comparison between different models and to observations.*

4. Line81: Do xe and xw have to be land points in the code? Is possible to calculate transports between water points using the current version?
   *It is possible to calculate transports between water points, however results should be viewed with caution as water will also pass to the left/right of the defined strait and the exact position of currents in the models is not known. Therefore, the code produces the requested transports, but it also gives a warning saying '!!!ATTENTION!!!: first/last point water, recheck indices line!'*
   *We added the following:*
   *The boundaries z_b, x_1, x_2 should be chosen such that no water can "escape" the desired coast-to-coast section. This can be ensured if x_e and x_w are land points and the auxiliary fields describing model ocean depths are used appropriately. **It is also possible to calculate transports between two water points, however***

*results should be viewed with caution and their correctness is left to the discretion of the user as water might bypass the strait.*

5. In Figure 2b, four grid cells with blue, green and yellow arrows are shown. The one on the upper-left corner is misleading because the red reference line does not touch it.
   We added a cell not touching the reference line, as it is also needed for the bilinear interpolation of the orthogonal transports onto the strait.

   We clarified it in the text at L116-118:
   *For every grid cell touching the strait **and their neighboring cells (needed for the interpolation onto the strait)** we calculate direction vectors of the u and v components (blue and green arrows), and normal vectors pointing from the tracer grid cell in the direction of the strait (yellow arrows).*

   And at L123-125:
   *Using the magnitudes of the projection vectors we calculate the u and v components pointing orthogonally onto the strait at all grid cells touching the strait **and their neighboring cells (needed for the bilinear interpolation).** Then, we multiply them with the respective vertical cell thicknesses at the u/v points and interpolate those orthogonal "transports" bilinearly onto the closest points on the reference line (black crosses in Fig. 2b, called T_proj henceforth). In a final step we divide by the vertical thickness of the cells on the reference line to obtain velocities again.*

6. Line155: The authors said "the section can be kinked". Here does "kinked" mean a zigzag line? If it does, then this is good because we do need zigzag sections from time to time. According to the definition of *def_indicies()* function, this is only possible if *set_latlon = True* and *lon_p* and *lat_p* are provided. Maybe you can put it more clear in the paper or in the code documentation.
   Yes that's true, at the moment that's the only way to define zigzag sections. We added the following:
   **The determination of section positions for transport calculations is accomplished in the def_indices function. Users can specify the start and end points of a section using the 'coords' parameter in the 'transports' function, the section will then follow the shortest distance along the sphere. Alternatively, users may pass specific coordinates by setting the 'set_latlon=True' parameter and providing a list of latitude ('lat_p') and longitude ('lon_p') points. The latter option also allows the calculation of "kinked" sections.**

7. Line157: a reference line "consisting of equally spaced latitude-longitude pairs". What is the interval between points on the reference line? (about 0.1deg according to *def_indices()*, but why?)
   We chose 0.1 deg to ensure that also the higher resolution (0.25°x0.25°) models don't skip any points. This works great for models with a resolution of

up to approx. 0.25°x0.25° (lower resolution models produce duplicate indices, however those are removed automatically), for even higher resolution models we advise to use an even denser spacing. **We adapted the code for the newest version of StraitFlux, so that the resolution is checked automatically, and the interval is adapted accordingly to 0.4*resolution** (so about 0.4 for 1° models and 0.1 for 0.25° models and even higher for higher resolution models). We added the following to the manuscript: *Using the 'coords' option* *the function generates a reference line (ref_line) consisting of equally spaced latitude-longitude pairs* ***whereat the interval between points on the reference line is set automatically to suite the resolution of the model. When passing coordinates via the 'set_latlon=True' option we advise the user to use intervals not larger than 0.4 times the resolution of the model (e.g., intervals of 0.1° for models with a resolution of about 0.25°) as coarser intervals might lead to the skipping of grid points and generate broken lines. This might create duplicates in the indices found, however those will be removed automatically.***

8. Line165: when I read the manuscript, I thought that if the interval between two neighboring points on the reference line is very small, then there should be duplicates in the indices found by *select_points()*. And when I read the code of *check_availability_indices()*, I realized that duplicates will be removed by the code automatically. But I still think it will be better if the authors explicitly write out that "**there might exist duplicates in the indices found by** *select_points()*, **but the code will later on remove the duplicates automatically**". This helps reader like me to release the puzzles.
   See answer above

***9.*** Line170: the first and last point should be land grid points. Again, is it possible to calculate transports between two ocean grid points?
   Yes, it is possible, although we would advise to do it with caution. We adapted the sentence to the following:
   *To prevent water from "escaping",* ***we advise the user to place the first and last point of the defined section over land. Transports may also be calculated for sections between two ocean points, however a warning will be given to the user as those should be treated with caution.***

10. Line175-176: left/right/above/below

It is straightforward to use these words to depict the directions of the grid line, but cautions need to be taken where grid lines are distorted greatly in the Arctic Ocean. For example, in a tripolar grid, if we draw a section following a meridional grid line in the Arctic Ocean (e.g., along the Lomonosov ridge), is the cell coming from left/right or above/below to its next one?
Note that the orientation of left/right/above/below is defined concerning the native grid indices (x,y), therefore the local direction of x is not necessarily west-east. E.g. coming from left in this sense means coming from x-1 to x. This should distinctly assign each direction (coming from above/below/left/right) to one move along the

native gridlines (coming from $y_{i+1}/y_{i-1}/x_{i-1}/x_{i+1}$), even for straits in the far north (see indices of Lomonosov on tripolar grid below).

[Figure]

We adapted Fig. 3 to show the indices selection process in respect to the native grid lines (x and y instead of lat/lon on the figure axes). Latitude and Longitude lines are shown in grey. Further, we changed the grid cell indices in Fig. 3c) to j to avoid confusion with the indices along the strait i in 3a+b).

[Figure]

*Figure 3. Illustration of the indices selection process using select_points.* **Lines of constant latitude/longitude are shown in grey.** *a) and b) Determination of consecutive grid-cells* **on the native grid** *by comparing distances d (orange lines) of 4 surrounding grid-cells* **for all equally spaced points i along the reference line.** *c) Specification whether a u or v component should be taken.*

We adapted the indexing for Fig3c) in the text (L176-178):
*For instance, to get to cell **j** in Fig. 3c, we came from left, hence the v component of cell **j** is taken. In order to get to cell **j** + 1 we come from below, therefore the u component of cell **j** + 1 is taken.*

And we added the following footnote:
**Note that we follow the native grid points (x,y) and the local direction of x is not necessarily west-east and the local direction of y not south-north. For instance, coming from left means coming from point $[x_{i-1},y_i]$ to point $[x_i,y_i]$.**

1. Line227: similar to Line165: Are duplicates found by the code removed automatically?
   Yes, they are. We adapted the section to the following:
   *As for the LM, the first step is to find the closest points on the native grids to the reference line.* **The selection of the indices proceeds similar as for LM, however herein additionally to the closest points to the reference line also the four immediate neighboring cells of the closest points are used. Those are needed for the interpolation of the transports onto the reference line. Again, any duplicate indices are removed automatically.**

2. Line313: typo "VPN".
   We corrected it.

3. Line339: ITF needs to be clarified.
   We clarified it as Indonesian Throughflow Region.

4. Line373: what is "i.a."?
   We changes it to "for instance".

5. Line392: I can see that on github, there is an instruction on how to install StraitFlux. My own experience, however, is that I can shoot myself in the foot if I mix the use of conda and pip. In consideration of the future update of StraitFlux (e.g., Line128-129), I strongly encourage the authors to upload this tool to conda-forge.
   Thank you for the idea, we will consider uploading StraitFlux to conda-forge for a future version.

6. Line397: typo "downlaoded".
   We corrected the typo.

**Code bugs:**

I test the code by myself and I do encounter some errors which turn out to be caused by bug in the code. When I ran the Examples.ipynb script, I got the following error.

TypeError: check_Arakawa() takes 4 positional arguments but 5 were given.

Then I found that *check_Arakawa()* only accepts 4 rather than 5 positional arguments. So in line113 and line131 of *mastersciprt_line.py*, there should be no the **product** parameter.
Thank you for checking the code. We corrected the error in the newest version of StraitFlux.

---

## Author Response (AR2)

We thank Shizhu Wang for another round of constructive reviews. The revisions are provided below.

The authors of Winkelbauer et al. present a thoroughly revised version of their manuscript. Their responses to my comments on the first version are both comprehensive and considerate. They have addressed all my key concerns. In particular, adding figures on dealing with overlapping cyclic boundary points is very helpful. I agree that this manuscript be published after some minor modification.

Line134: in a final step we divide WHAT by …

*Then, we multiply them with the respective vertical cell thicknesses at the u/v points and interpolate those orthogonal "transports" bilinearly onto the closest points on the reference line (black crosses in Fig. 2b, called Tproj henceforth). In a final step we divide* **the interpolated "transports"** *by the vertical thickness of the cells on the reference line to obtain velocities again.*

Line187, and according to your response to my minor comment 4 and 9: it would be better to place the first and last points over land to prevent water from "escaping". What do you mean by "escaping" here? To my understanding, if I put the starting point over land and the last point in the middle of a strait, then the code should compute the results of half of the strait. Am I right?

Yes, it would calculate transport for half of the strait. However, as positions of currents are not the same for different models/reanalyses, it's possible that significant parts of the currents pass the strait for some models, while they are integrated for others. This may yield to significantly different results when looking at integrated transports, which might be prevented when placing the strait from land to land. However, we admit that calculation of pure oceanic straits may still be useful, especially when considering cross-sections. Therefore, we changed the wording a bit:

*This is done for all points on the reference line and results in a closed sequence of grid cells along the reference line (filled blue cells). To prevent water from* **flowing around the strait**, *we advise the user to place the first and last point of the defined section over land.* **Transports may also be calculated for sections between ocean points. However, as the position of currents might differ between models this could lead to currents circulating around the strait and result in significantly different results. Therefore, should the strait start/end in water a warning will be given to the user and transports should be treated with caution.** *The user is provided with figures of the selected line and the model land-sea mask, which can be used to check the position and length of the desired strait.*

Line324: we are talking about ocean current not "wind component" here.

*Regridding would involve rotating the* **ocean velocity** *components to the new* **flow** *direction (eastward/northward) prior to the interpolation as done e.g. by He et al. (2019).*